

# Activities concentration of radiocesium in wild mushroom collected in Ukraine 30 years after the Chernobyl power plant accident

Makiko Orita[1], Yuko Kimura[2], Yasuyuki Taira[1], Toshiki Fukuda[1], Jumpei Takahashi[3], Oleksandr Gutevych[4], Serghii Chornyi[4], Takashi Kudo[5], Shunichi Yamashita[6] and Noboru Takamura[1]

[1] Department of Global Health, Medicine and Welfare, Atomic Bomb Disease Institute, Nagasaki University, Nagasaki, Japan
[2] Department of Health and Welfare, Kawauchi Municipal Government, Fukushima, Japan
[3] Center for International Collaborative Research, Nagasaki University, Nagasaki, Japan
[4] Zhitomir Inter-Area Medical Diagnostic Center, Korosten, Ukraine
[5] Department of Radioisotope Medicine, Atomic Bomb Disease Institute, Nagasaki University, Nagasaki, Japan
[6] Department of Radiation Medical Sciences, Atomic Bomb Disease Institute, Nagasaki University, Nagasaki, Japan

## ABSTRACT

Mushrooms are recognized as one of the main contributors to internal radiation exposure from the activity concentration of radiocesium released by the accident at the Chernobyl nuclear power plant (CNNP). We evaluated the activity concentrations of the artificial radionuclides (radiocesium) in wild mushrooms collected in 2015 from Korosten and Lugine, Zhitomir region, Ukraine, located 120 km away from the CNPP. Cesium-137 was detected in 110 of 127 mushroom samples (86.6%). Based on the average mushroom consumption (5 kg per year), we calculated committed effective doses ranging from 0.001–0.12 mSv. Cesium-137 remains in the wild mushrooms even 30 years after the accident, but the committed effective doses are limited by the amount of contaminated mushrooms consumed. However, evaluation of internal radiation exposure and assessment of environmental radioactivity in the surrounding area affected by the nuclear accident are still necessary in order to relieve anxiety about internal radiation exposure, as long as the possibility of consumption of contaminated mushrooms remains.

## INTRODUCTION

The accident at the Chernobyl Nuclear Power Plant (CNPP) occurred in April 1986. This accident released huge amounts of radionuclides, including radioiodine and radiocesium, into the environment, contaminating the lands of Ukraine, the Republic of Belarus, and the Russian Federation (*Taira et al., 2011*). Cases of hot spot contamination with radiocesium and high levels in mushrooms have also been recorded in nearby countries such as Poland, Sweden, and Norway, while less elsewhere in Europe (*Bakken & Olsen,*

Corresponding author
Makiko Orita, orita@nagasaki-u.ac.jp

*1990*; *Cocchi et al., 2017*; *Falandysz & Borovička, 2013*; *Falandysz et al., 2015*; *Falandysz et al., 2016*; *Strandberg, 2004*; *Zalewska, Cocchi & Falandysz, 2016*). More than 30 years have passed since the accident, but cesium-137 ($^{137}$Cs) remains a radioactive nuclide of interest due to its relatively long half-life (30 a) (*International Atomic Energy Agency, 2006*). Soil contamination by $^{137}$Cs led to the contamination of locally produced foods and resulted in internal exposure of the residents.

Several studies have reported a relationship between whole body exposure, the activity concentration of $^{137}$Cs, and the activity concentration in local foods. Radiocesium is known to concentrate in wild mushrooms (*Hoshi et al., 2000*; *Travnikova et al., 2001*; *Hoshi et al., 1994*; *Kaduka et al., 2006*; *Smith, Taylor & Sharma, 1993*; *Mukhopadhyay et al., 2007*) and, as mentioned, the most contaminated originated from the regions of Ukraine, Gomel in Belarus, and countries north and west of Chernobyl (*Bakken & Olsen, 1990*; *Bulko et al., 2014*; *Falandysz et al., 2015*; *Grodzinskaya et al., 2003*). *Hoshi et al. (2000)* reported that children residing near Chernobyl who consumed mushrooms showed a high $^{137}$Cs body burden, suggesting that mushrooms are one of the main contributors to internal radiation exposure from the radiocesium released following nuclear disasters.

The Great East Japan Earthquake of March 2011 and the resulting tsunami triggered a nuclear reactor accident at the Fukushima Daiichi Nuclear Power Station (FDNPS). This accident raised concerns among residents regarding the risks of internal exposure through the consumption of locally produced foods, especially edible wild plants and wild mushrooms, which are traditional parts of the regular diet (*Orita et al., 2016*). We have evaluated the activity concentration of radiocesium in wild mushrooms collected in Kawauchi village, one of the areas affected by the Fukushima accident, and we detected radioactive cesium exceeding 100 Bq/kg (the current regulatory value for radiocesium in foods in Japan) in 125 of 154 mushroom samples (81.2%) in 2013 and in 147 of 159 mushroom samples (92.4%) in 2015 (*Nakashima et al., 2015*; *Orita et al., 2017*).

A comprehensive radiological protection assessment needs to be implemented over the long term for the recovery of the Chernobyl and Fukushima regions; however, follow-up evaluations of radiocesium concentrations in wild mushrooms in the areas around the Chernobyl have not been conducted. In this study, we evaluated the current activity concentration of radiocesium in wild mushrooms collected in Ukraine in order to estimate the potential internal radiation exposure of the area residents thirty years after the accident.

## MATERIALS & METHODS

All wild mushrooms were collected in Korosten and Lugine, Zhitomir region, Ukraine. This area is located 120 km southwest of the CNPP and was heavily affected by the accident. In Ukraine, the contamination with radionuclides was severe in the regions to the northwest and west of the power plant site. In addition, a classification of four contamination zones was established in Ukraine. These zones were defined according to their soil contamination levels of $^{137}$Cs as 'Zone 1' (>1,480 kBq/m$^2$), 'Zone 2' (555–1,480 kBq/m$^2$), 'Zone 3' (185–555 kBq/m$^2$), and 'Zone 4' (37–185 kBq/m$^2$) (*Kimura et al., 2015*).

In September to November 2015, we collected 127 mushrooms from three species: 36 samples of *B. edulis*, 42 samples of *L. aurantiacum*, and 49 samples of *L. scabrum*. Among

the different zones, 51 samples were collected in Zone 2, 38 in Zone 3, and 38 in Zone 4. Data for the radiocesium concentrations in mushrooms from Zone 1 were not available.

After collection, all samples (approximately 200 g each fresh weight) were washed with water to remove soil and then dried using a heater (65 °C for 24 h, 105 °C for 1 h). The samples were crushed to a powder with a mortar. The samples (approximately 0.017 kg each dry weight) were enclosed in 100-mL polypropylene containers, and analyzed for 3,600 s with a high-purity germanium detector (ORTEC®, GMX30-70; Ortec International Inc., Oak Ridge, TN, USA) coupled with a multi-channel analyzer (MCA7600; Seiko EG&G Co., Ltd., Chiba, Japan).

The measurement time was set to detect the objective radionuclide, and the gamma-ray peak was 661.64 keV for $^{137}$Cs (30 a). Decay corrections were made based on the sampling date, and the detector efficiency was calibrated for different measurement geometries using mixed-activity standard volume sources (Japan Radioisotope Association, Tokyo, Japan). The concentrations of radiocesium were automatically adjusted for the date of collection, and the data were defined as the concentration at the collection date. The detection limit was 28.5 Bq/kg for $^{137}$Cs (median, for dry mushrooms).

We calculated the ratio of the weight of dried and raw mushrooms and obtained the following formula:

$$\left\{ \text{Radiocesium concentrations in raw mushrooms}\left(\frac{\text{Bq}}{\text{kg}}\right)\right\} = \left\{\frac{(\text{Raw weight})}{(\text{Dried weight})}(\text{kg})\right\}$$
$$\times \left\{\text{Radiocesium concentration in dried mushrooms}\left(\frac{\text{Bq}}{\text{kg}}\right)\right\}. \qquad (1)$$

In this article, we have presented the concentrations for raw mushrooms, and we used these in our analysis.

The committed effective dose based on mushroom concentration intake was calculated using the following formula:

$$H_{\text{int}} = C \cdot D_{\text{int}} \cdot e$$

where $C$ is the activity concentration of the detected artificial radionuclide (radiocesium) (Bq/kg), $D_{\text{int}}$ is the dose conversion coefficient for adult intake (age 20 and older, $1.3 \times 10^{-2}$ µSv/Bq for $^{137}$Cs) (*International Commission on Radiological Protection, 1996*), and $e$ is the daily intake value (age 20 and older, 5 kg/year, the average intake of Russian citizens) (*Škrkal et al., 2013*; *Malátová& Tecl, 2001*). We assumed that similar annual intakes could be attributed to each species.

Data are expressed as medians, minimums, and maximums. Differences in the concentrations of radiocesium in species of mushroom and at each sampling site were evaluated using Analysis of Variance (ANOVA). Probability values less than 0.05 were considered statistically significant. All statistical analysis was performed using SPSS statistics 22.0 (SPSS Japan, Tokyo, Japan).

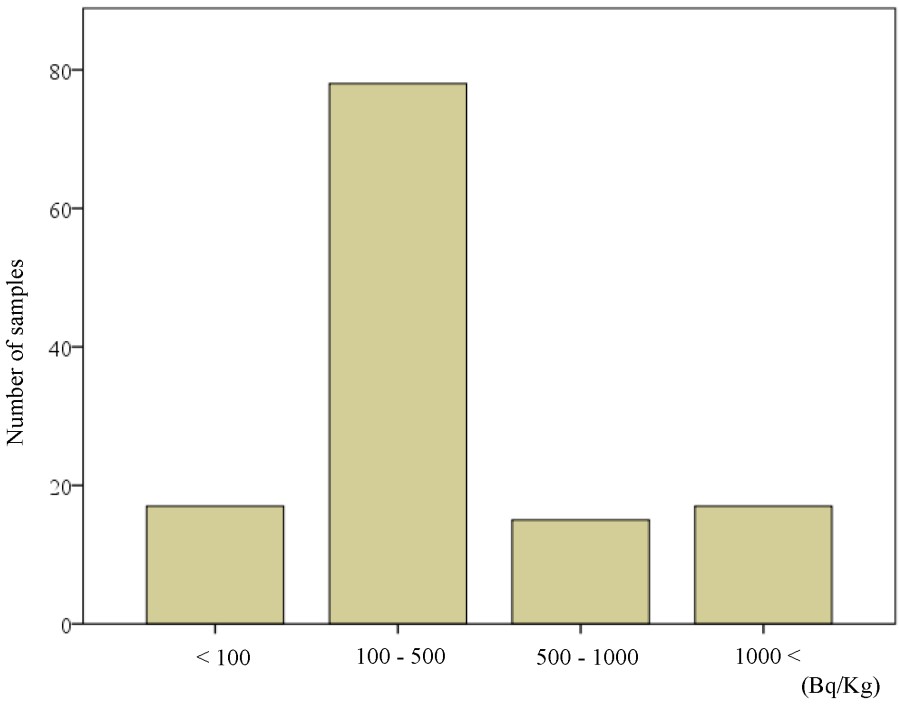

**Figure 1** Distribution of activity concentration of Cesium-137 in raw mushrooms collected in 2015 from the Chernobyl area of Korosten and Lugine, Ukraine.

**Table 1** Concentration of radiocesium in mushrooms collected in 2015 from the Chernobyl area of Korosten and Lugine, Zhitomir region, Ukraine.

| Type | Number | $^{137}$Cs—raw median (min–max) (Bq/Kg) |
|---|---|---|
| *Boletus edulis* | 36 | 580 (27–1,800) |
| *Leccinum aurantiacum* | 42 | 250 (15–480) |
| *Leccinum scabrum* | 49 | 290 (18–1,400) |

## RESULTS

Among the 127 mushroom samples collected, 17 mushroom samples (13.4 %) had no detectable levels of $^{137}$Cs; however, 77 mushroom samples (60.6 %) had radiocesium levels of 100–500 Bq/kg, 16 (12.6 %) had levels of 501–1,000 Bq/kg, and 17 (13.4 %) had levels >1,000 Bq/kg (Fig. 1).

The concentration of radiocesium in each mushroom species is shown in Table 1. The median radiocesium concentration of *Boletus edulis* was 580 Bq/kg, with a minimum and maximum of 2 and 1,800 Bq/kg, respectively. The median radiocesium concentration of *Leccinum aurantiacum* was 250 Bq/kg, with a minimum and maximum of 15 and 480 Bq/kg, respectively. The median radiocesium concentration of *L. scabrum* was 290 Bq/kg, with a minimum and maximum of 18 and 1,400 Bq/kg, respectively. Comparison of the concentration of radiocesium in each species revealed no difference between *L. aurantiacum*

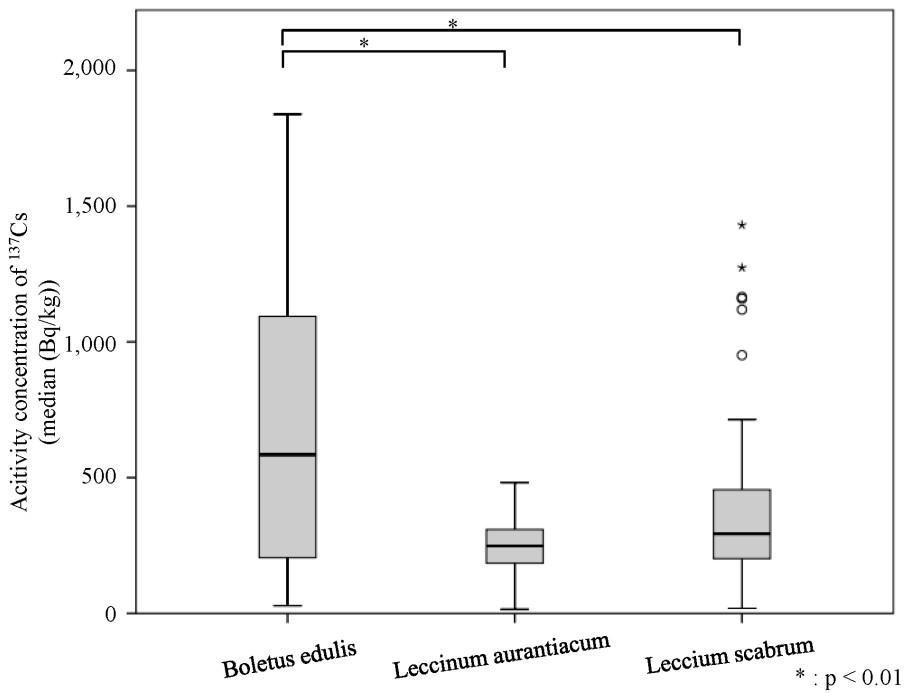

**Figure 2 Activity concentration of Cesium-137 in different species of raw mushrooms collected in 2015 from the Chernobyl area of Korosten and Lugine, Ukraine.**

and *L. scabrum* in terms of radiocesium concentration, whereas the levels were significantly higher in *B. edulis* than in *L. aurantiacum* ($p < 0.01$) and *L. scabrum* ($p < 0.01$) (Fig. 2). Conversely, no differences were detected in radiocesium concentration among the different zones.

Among the 110 mushrooms collected that contained radiocesium, the calculated committed effective doses would have ranged from 0.001–0.12 mSv if the residents had ingested the contaminated mushrooms over a one-year period.

## DISCUSSION

The most important radionuclides released following the nuclear disasters at Chernobyl and Fukushima, from the perspective of the general population, were iodine-131 ($^{131}$I), cesium-134 ($^{134}$Cs), and $^{137}$Cs. Especially $^{137}$Cs remains the main factor for internal radiation exposure of the area residents because it has a much longer half-life of 30 a (*Sekitani et al., 2010*; *International Atomic Energy Agency, 2006*).

Since the Chernobyl accident, various studies have been conducted to clarify the influence of the activity concentration of radiocesium in forest-derived products, including mushrooms. For example, *Travnikova et al. (2001)* evaluated the average content of radiocesium in mushrooms collected in Veprin village, Russia, from 1994 to 1998, and found average concentrations of $^{137}$Cs of 14,500, 2,550 and 8,980 Bq/kg in *Boletus luteus*, *B. chanteral*, and *B. russula*, respectively. In the present study, $^{137}$Cs was detected in 110 of 127

(87%) wild mushrooms collected in Ukraine, suggesting a high frequency of occurrence of $^{137}$Cs in the local mushrooms even 30 years after the accident.

The radiocesium concentrations in wild mushrooms showed no differences in the different zones, although a significantly higher activity concentration of radiocesium was noted in *B. edulis* than in *L. aurantiacum* or *L. scabrum*. We recently collected mushroom samples in 2013 and 2015 in Kawauchi village, Fukushima, one of the areas affected by the Fukushima accident, to evaluate the activity concentration of radiocesium, and found that the proportion and concentration of radiocesium might depend on the species of mushroom (*Nakashima et al., 2015*; *Orita et al., 2017*). *Yoshida & Muramatsu (1994)* reported that the mycelium habitat seemed to be one of the factors determining the radiocesium concentration in mushrooms. On the other hand, we found no relationship between habitat and radiocesium concentration in this study. Further studies are required to clarify the factors that determine the radiocesium concentration, considering the species of mushroom, mycelium habitat and environmental radioactivity in the place where mushrooms are collected.

We calculated committed effective doses ranging from 0.001–0.12 mSv, based on the average annual consumption of mushrooms. After the Chernobyl accident, *Hoshi et al. (2000)* conducted measurements of whole - body counter of the $^{137}$Cs from 1991 to 1996 for children in Bryansk Oblast, Russia, which is affected by contamination following the Chernobyl accident. They found that the most common food items contributing to $^{137}$Cs intake in children were mushrooms, wild vegetables, and wild beers. *Sekitani et al. (2010)* also evaluated the whole - body counter of $^{137}$Cs in residents of Bryansk Oblast, Russian Federation from 1998 to 2008, and found that $^{137}$Cs concentration was significantly higher in autumn than in other seasons due mainly to an increased intake of forest products, such as mushrooms, in autumn. These results suggest that residents might have consumed contaminated forest products. By contrast, we found that the committed effective doses were limited, even if the residents consumed contaminated mushrooms. However, evaluation of internal radiation exposure and assessment of environmental radioactivity remain necessary in the surrounding areas affected by the nuclear accident in order to relieve the residents' anxiety about internal radiation exposure.

Our study has several limitations. First, we could not evaluate the relationship between radiocesium concentrations in mushrooms and the concentrations in the soil. Second, additional analytical uncertainties arose because the committed effective doses from dietary intake of mushrooms cannot measure the day-to-day variations in individuals in Ukraine. Household treatment can have a high impact on the content of metallic elements and radionuclides in cooked mushroom, and typically, a decrease of activity concentration can be expected (*Drewnowska et al., 2017*; *Steinhauser & Steinhauser, 2016*), however, this aspect was not considered in our assessment.

## CONCLUSION

We evaluated the activity concentration of $^{137}$Cs in wild mushrooms collected in Ukraine in 2015, and found that $^{137}$Cs still remains in wild mushrooms even 30 years after the accident.

The committed effective doses are limited by the amounts of contaminated mushrooms consumed; however, we believe that a long-term comprehensive risk evaluation, including measurements of the activity concentration of radiocesium in locally produced foods, such as mushrooms, is necessary for recovery from the Chernobyl nuclear disaster.

### Funding

The authors received no funding for this work.

### Competing Interests

The authors declare there are no competing interests.

### Author Contributions

- Makiko Orita conceived and designed the experiments, performed the experiments, analyzed the data, wrote the paper, prepared figures and/or tables.
- Yuko Kimura conceived and designed the experiments, performed the experiments, analyzed the data, wrote the paper.
- Yasuyuki Taira performed the experiments, analyzed the data.
- Toshiki Fukuda performed the experiments, analyzed the data, prepared figures and/or tables.
- Jumpei Takahashi, Oleksandr Gutevych and Serghii Chornyi performed the experiments.
- Takashi Kudo and Shunichi Yamashita wrote the paper.
- Noboru Takamura analyzed the data, contributed reagents/materials/analysis tools, wrote the paper, reviewed drafts of the paper.

### Data Availability

The raw data has been provided as a Supplemental File.

### Supplemental Information

Supplemental information for this article can be found online at http://dx.doi.org/10.7717/peerj.4222#supplemental-information.

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
