# Peer review of "Activities concentration of radiocesium in wild mushroom collected in Ukraine 30 years after the Chernobyl power plant accident"

_PeerJ, doi:10.7717/peerj.4222_

## Round 0.1 · original submission · Minor Revisions

I agree with the reviewers in that you need to cite more pertinent literature. Reviewer 1 especially gave you very good suggestions you should respond to.

Reviewer 1 ·

Basic reporting

Comment to manuscript : #20396
Activities concentration of radiocesium in wild mushroom collected in Ukraine after the 30 years from the Chernobyl Power Plant Accident (#20396)

General comment:
Useful data; can be accepted after some upgrade (also with a recent literature)

Suggested improvements:


Is: This accident released huge amounts of radionuclides, including radioiodine and radiocesium, into the environment, contaminating the lands of Ukraine, the Republic of Belarus, and the Russian Federation (Taira Y et al., 2011).
Suggested correction: This accident released huge amounts of radionuclides, including radioiodine and radiocesium, into the environment, contaminating the lands of Ukraine, the Republic of Belarus, and the Russian Federation (Taira Y et al., 2011). A cases of a hot spot contamination with radiocaesium and high levels in mushrooms has been recorded also in the nearby countries such as Poland, Sweden or Norway while less elsewhere in Europe (Bakken and Olsen 1990; Cocchi et al. 2017; Falandysz and Borovička 2013; Falandysz et al. 2015, 2016; Strandberg, 2004; Zalewska et al. 2016)
Examplum gratia:
[ Bakken and Olsen, 1990, Accumulation of radiocaesium in fungi. Can J Microbiol 36:704-710
Cocchi et al. , 2017, Radioactive caesium (134Cs and 137Cs) in mushrooms of the genus Boletus from the Reggio Emilia in Italy and Pomerania in Poland. Isotopes Environ Health Stud 53:doi.org/10.1080/10256016.2017.1337761
Falandysz and Borovička. 2013, Macro and trace mineral constituents and radionuclides in mushrooms – health benefits and risks. Appl. Microbiol. Biotechnol. 97, 477-501 (2013).
Falandysz et al. 2015, Evaluation of the radioactive contamination in Fungi genus Boletus in the region of Europe and Yunnan Province in China.. Appl. Microbiol. Biotechnol. 99, 8217-8224.
Falandysz et al. 2016, Determination of activity concentration of 137Cs and 40K in some Chanterelle mushrooms in Poland and China J Environ. Sci. Poll. Res. 23, 20039-20048.
Strandberg, 2004, Long-term trends in the uptake of radiocesium in Rozites caperatus. Sci. Total Environ. 327, 315-321.
Zalewska et al. , 2016, Radiocaesium in Cortinarius spp. mushrooms in the regions of the Reggio Emilia in Italy and Pomerania in Poland. Environ. Sci. Poll. Res. 23, 23169–23174.]


Is: Radiocesium is known to concentrate in wild mushrooms (Hoshi et al., 2000; Travnikova et al., 2001; Hoshi et al., 1994; Kakuda et al., 2006; Smith et al., 1993; Bulko et al., 2014).

Suggested: Radiocesium is known to concentrate in wild mushrooms (Hoshi et al., 2000; Travnikova et al., 2001; Hoshi et al., 1994; Kakuda et al., 2006; Smith et al., 1993) and, as mentioned, the most contaminated originated from the regions of Ukraine, Gomel in Belarus or a countries north and west of Chernobyl (Bakken and Olsen 1990; Bulko et al., 2014; Falandysz et al. 2015; Grodzinskaya et al. 2003 and 2013).

A.A. Grodzinskaya, M. Berreck, K. Haselwandter, S.P. Wasser. Radiocesium contamination of wild-growing medicinal mushrooms in Ukraine. Int. J. Med. Mush. 5, 61-86 (2003).

A.A. Grodzinskaya, S. A. Syrchin, N.D. Kuchma, S.P. Wasser. Macromycetes accumulative activity  in radionuclide contamination conditions of the Ukraine territory. Part 6. - P.217-260, 368-373. In: Mycobiota of Ukrainian Polesie: Consequences of the Chernobyl disaster. Kiev: Naukova dumka, 2013. (In Russian).


Is: ……. locally produced foods, especially edible wild plants and wild mushrooms, which are traditional parts of the regular diet (Orita et al., 2016).

Suggested: ……. locally produced foods, especially edible wild plants and wild mushrooms, which are traditional parts of the regular diet (Orita et al., 2016).


Is: …. radiocesium in each type of mushroom ..
Suggestion: … type? What do you mean? (should be “species of mushroom”)?

Is: … (p<0.01) and L. scabrum (p<0.01) ……
Proper record is: (p < 0.01) and L. scabrum (p < 0.01) …..


Is: Our study has several limitations. First, we could not evaluate the relationship between
radiocesium concentrations in mushrooms and the concentrations in in the soil. Second, additional analytical uncertainties arose because the committed effective doses from dietary intake of mushrooms cannot measure the day-to-day variations in individuals in Ukraine.
Suggested: Our study has several limitations. First, we could not evaluate the relationship between radiocesium concentrations in mushrooms and the concentrations in in the soil. Second, additional analytical uncertainties arose because the committed effective doses from dietary intake of mushrooms cannot measure the day-to-day variations in individuals in Ukraine. A household treatment can have a high impact on content of metallic elements and radionuclides in cooked mushroom and usually a decrease of activity concentration can be expected (Drewnowska et al. 2017; Steinhauser and Steinhauser, 2016)) but this aspect was not considered in our assessment.
Drewnowska et al., 2017. Leaching of arsenic and sixteen metallic elements from Amanita fulva mushroom. LWT - Food Sci technol 84:861–866
Steinhauser and Steinhauser, 2016, A simple and rapid method for reducing radiocesium concentrations in wild mushrooms (Cantharellus and Boletus) in the course of cooking. J Food Prot 79:1995–1999

Other remarks:
All results (data) have to be rounded and only two significant figures have to be shown!!!
Not: 583 (27–1,838)
But: 580 (27–1,800)

It also matters as relates to assessed exposure (mSv).

Experimental design

Is Ok.

Validity of the findings

Good

Additional comments

Useful data; can be accepted after some upgrade (also with a recent literature); look into a detail comment

Reviewer 2 ·

Basic reporting

There are many important literature references missing in the text regarding radiocaesium concentration in mushrooms in areas affected by Chernobyl accident. For example, there is no mention to the ban of mushroom collection, and how it affected the internal dose due to their consumption. Raw data are shared. A map showing sampling points is missing.

Experimental design

The analysis of the radiocaesium concentration in mushrooms in Ukraine and how it may be related to those collected after Fukushima accident is an interesting topic. However, there are only data for three species of mushrooms in Ukraine and none for Japan. The authors use raw and dried mushrooms, this is quite confusing, since a dried mushrooms is not cooked. It would be better to express this in terms of fresh and dry mass. It is not clear if the samples were measured dried or not.

Validity of the findings

No comment

Additional comments

The novelty and the importance of the manuscript is very low, as it only gives the mean value of three species of mushrooms from Ukraine. There is no comparison with data from mushrooms collected just after Chernobyl accident. There are many significant references missing. The authors only focused on previous work done mainly by Japanese scientist in that area. In my opinion, this manuscript show only that the authors collected a lot of mushrooms samples and measured them. Therefore my recommendation is rejection.

·

Basic reporting

The article is nicely written, clear and lucid English. Sufficient literature citation. I just mention one more article in line 52, 53 of introduction part. Six references have been cited. The following reference, which is newer than some of the reference cited also strongly advocates your comment in the relevant section. Therefore may be included:

B Mukhopadhyay et al., J. Radioanal. Nucl. Chem. 273 (2007) 415-418.

Experimental design

Research question is well defined, relevant and meaningful. Methods described with sufficient detailing.

According to APS, now year is abbreviated as 'a' (annum), when half life of a radionuclide is described. Please change '30 y' to '30 a' while mentioning half-life of Cs-137.

In line 94 you described the detection limit of 137-Cs 28.5 Bq/kg (as you have calculated from median value). But it must be much less for HPGe detector for 137 Cs. I think you should calculate detection limit of 137-Cs from the background.

You have not mentioned how much dry mushroom was taken for measurement of 137Cs and in what geometry. This is important information for gamma spectrometric measurement.

Validity of the findings

Data is robust, statistically sound and controlled. I would be nice if you
(i) describe any significant physical variation of the mushrooms from contaminated regions and non-contaminate regions taken from anywhere of the world.

(ii) In figure 3 and 4, first data has very high error as calculated by you. Can you opine for this in discussion section.

Additional comments

As mentioned above.

---

## Round 0.2 · accepted · Accept

You have met the reviewers comments adequately.